# Decoupled Finetuning for Domain Generalizable Semantic Segmentation

**Jaehyun Pahk**[1]   **Donghyeon Kwon**[1]   **Seong Joon Oh**[3]   **Suha Kwak**[1,2]

Dept. of CSE, POSTECH[1]   GSAI, POSTECH[2]   Tübingen AI Center, Universität Tübingen[3]

## Abstract

Joint finetuning of a pretrained encoder and a randomly initialized decoder has been the de facto standard in semantic segmentation, but the vulnerability of this approach to domain shift has not been studied. We investigate the vulnerability issue of joint finetuning, and propose a novel finetuning framework called Decoupled FineTuning (DeFT) for domain generalization as a solution. DeFT operates in two stages. Its first stage warms up the decoder with the frozen, pretrained encoder so that the decoder learns task-relevant knowledge while the encoder preserves its generalizable features. In the second stage, it decouples finetuning of the encoder and decoder into two pathways, each of which concatenates an adaptive component (AC) and retentive component (RC); the encoder and decoder play different roles between AC and RC in different pathways. ACs are updated by gradients of the loss on the source domain, while RCs are updated by exponential moving average biased toward their initialization to retain their generalization capability. By the two separate optimization pathways with opposite AC-RC configurations, DeFT reduces the number of learnable parameters virtually, and decreases the distance between learned parameters and their initialization, leading to improved generalization capability. DeFT significantly outperformed existing methods in various domain shift scenarios, and its performance was further boosted by incorporating a simple distance regularization.

## 1   Introduction

The current de facto standard for learning semantic segmentation is to jointly finetune a pretrained encoder and a segmentation decoder on training data with segmentation labels (Long et al., 2015; Noh et al., 2015; Ronneberger et al., 2015; Chen et al., 2017; Zhao et al., 2017; Xie et al., 2021; Yu et al., 2022; Chen et al., 2023). This practice allows significant performance improvement, but it also often leads to models vulnerable to domain shift in testing caused by, for example, weather conditions and geolocations they do not experience in training (Ganin et al., 2016; Pan et al., 2018; Saito et al., 2018; Yue et al., 2019a; Choi et al., 2021). A straightforward solution to this issue is to collect a vast amount of training data from diverse domains. However, this does not guarantee that the collected data cover any potential test domains, and more importantly, pixel-wise class annotation for such data will be prohibitively expensive.

To resolve this issue, we study domain generalization for semantic segmentation, *i.e.*, learning a model on a single source domain so that it generalizes well to unseen, arbitrary target domains that may arise in testing (Yue et al., 2019b; Lee et al., 2022; Zhao et al., 2022; Chattopadhyay et al., 2023; Kim et al., 2023a). A large body of domain generalization research has focused on simulating diverse target domains by data or feature augmentation during training (Lee et al., 2022; Zhao et al., 2022; Chattopadhyay et al., 2023), or removing domain-specific information from features (Choi et al., 2021; Peng et al., 2022; Pan et al., 2018; 2019). Although these approaches have driven remarkable success, there is still large room for further improvement in that they do not take into account potential negative impact of the joint finetuning of encoder and decoder, the common practice in semantic segmentation, on domain generalization.

We argue that the joint finetuning of encoder and decoder can degrade the model's generalization capability. First, the joint finetuning causes the pretrained encoder to overfit to the source domain and thus corrupts its generalization capability (Kumar et al., 2022; Saito et al., 2023). Also, the

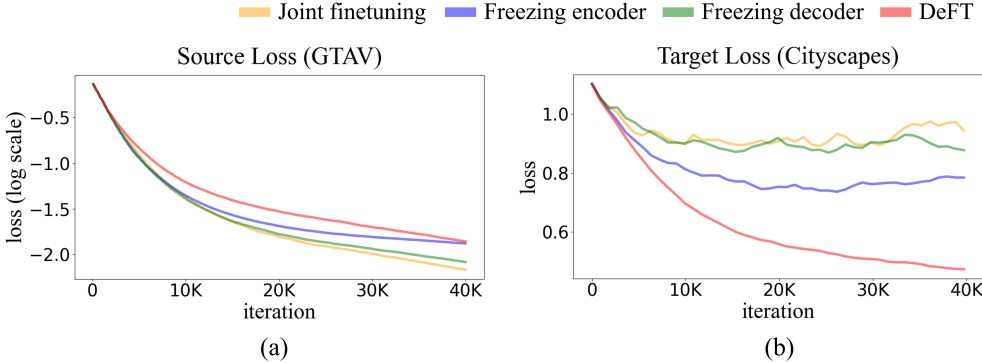

Figure 1: We empirically verify that the joint finetuning causes overfitting to the source domain and degrades the generalization capability by comparing (a) loss on the source domain–GTAV (Richter et al., 2016) and (b) that on an unseen target domain–Cityscapes (Cordts et al., 2016) during finetuning. Freezing either the encoder or decoder before being overfitted mitigates the issue to some extent, which suggests that preventing one of them from being trained with the other overfitted to the source domain may improve the entire model's generalization. Of course, this approach is far from the optimal solution due to the lack of task-relevant knowledge of the frozen module, as demonstrated in Table 1. Meanwhile, DeFT demonstrates significantly better generalization capability: its loss on the unseen target domain decreases more quickly and reliably during finetuning.

Table 1: Comparison between the proposed framework, DeFT, and freezing either an encoder or decoder. All the methods were trained on GTAV using ResNet-50 backbone. After the warm-up stage, we selectively froze either the encoder or decoder while continuing to update the other. The results indicate that the model cannot learn sufficient task-relevant knowledge when one module is not updated at all, especially the encoder, which contains most of the parameters of the model.

| Method | Cityscapes | BDD100K | Mapillary | Avg. |
|---|---|---|---|---|
| Freezing Encoder | 41.88 | 36.85 | 44.59 | 41.11 |
| Freezing Decoder | 42.52 | 38.75 | 45.03 | 42.10 |
| DeFT | **50.06** | **43.17** | **50.51** | **47.91** |

decoder relying on the encoder's output inevitably draws distorted decision boundaries, producing gradients that cause the encoder to be more overfitted. Figure 1 empirically verifies this argument.

Building on this insight, we propose a new, simple yet effective finetuning framework dubbed **De**coupled **F**ine**T**uning for domain generalization (DeFT). DeFT comprises two stages. In the first stage, the decoder is warmed up with a frozen pretrained encoder, following Kumar et al. (2022). By warming up on the source domain, the decoder learns the target task (*i.e.*, semantic segmentation in this paper) without distorting the generalizable knowledge of the pretrained encoder.

The main contribution of DeFT lies in its second stage, which finetunes both the encoder and decoder in a decoupled manner. Motivated by the observation in Figure 1, we propose decoupling the finetuning of the two trainable modules in the model, the encoder and decoder. To this end, we employ two parallel encoder-decoder pathways for finetuning, each combining an adaptive component (AC) and a retentive component (RC). AC is updated using the standard error backpropagation based on the training loss from the source domain, while RC is not updated using the gradients that might be overfitted to the source domain. As a result, RC maintains its generalization capability during finetuning, guiding the coupled AC's updates using its generalizable knowledge. Note that the encoder and decoder play different roles between AC and RC in different pathways.

A naïve strategy for managing RCs is not updating them at all, which however leads to a suboptimal solution due to the lack of task-relevant knowledge in RCs as demonstrated in Table 1. To allow RCs to learn task-relevant knowledge while preserving their generalization capability, we adopt a variant of exponential moving average (EMA) (Tarvainen & Valpola, 2017) that is biased towards the model's initial parameters, as an update scheme for RCs. This EMA method assigns higher

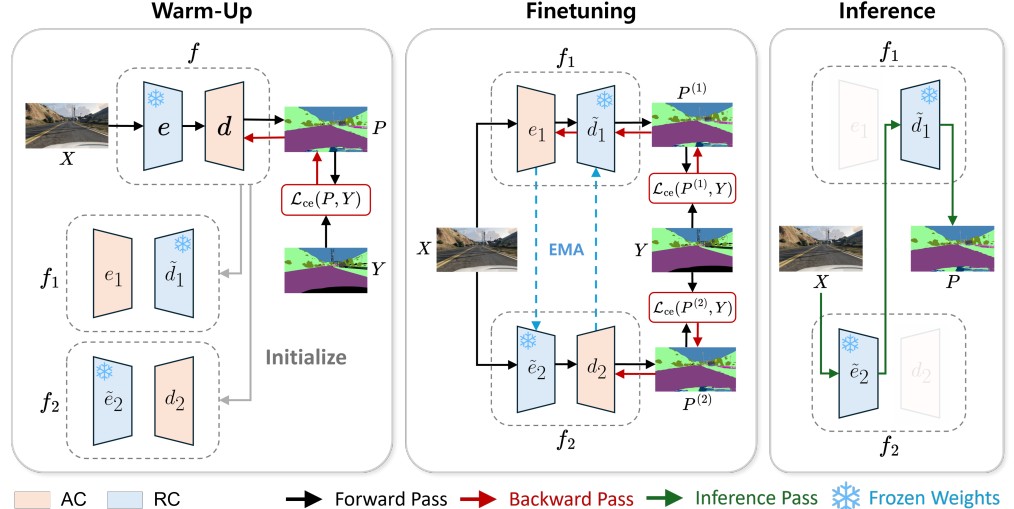

Figure 2: An overview of DeFT. The first step of DeFT is to warm up the decoder with the frozen, pretrained encoder. After warming up the decoder, the decoupled finetuning is conducted through two parallel encoder-decoder pathways. In the pathways, the parameters of encoders and decoders are initialized with those from the warmed-up model. In the second step, DeFT finetunes both the encoder and decoder in a decoupled manner: the RCs ($\tilde{d}_1$ and $\tilde{e}_2$) are updated by the exponential moving average of their counterpart ACs ($d_2$ and $e_1$), while the ACs are updated by gradients of the loss. Our final model for inference is configured as the combination of the well-generalized RCs ($\tilde{d}_1$ and $\tilde{e}_2$), i.e., $f_{\text{final}} = \tilde{d}_1 \circ \tilde{e}_2$.

weights to the early parameters during the finetuning, performing gradual temporal ensemble in the parameter space.

At the end of finetuning, DeFT produces two RC-AC pairs. Since RCs better preserve the rich and generalizable knowledge from pretraining and thus have better generalization capability than AC, we set our final model as the combination of two RCs, i.e., EMA encoder and EMA decoder. The overall pipeline of DeFT, including each stage, is illustrated in Figure 2.

Our method was evaluated on five different datasets, Cityscapes (Cordts et al., 2016), BDD-100K (Yu et al., 2020), Mapillary Neuhold et al. (2017), GTAV (Richter et al., 2016) and SYN-THIA (Ros et al., 2016), and it demonstrated superior performance to previous work in every experiment. In summary, our contribution is three-fold:

- We empirically demonstrate that joint finetuning of the encoder and decoder degrades generalization performance, and simply decoupling them in the finetuning process can substantially improve the performance.

- We propose a novel training framework for domain generalizable semantic segmentation, dubbed as DeFT, which finetunes the encoder and decoder in a decoupled manner. We also provide detailed analysis of our method through extensive experiments.

- DeFT was evaluated on various domain shift scenarios using multiple semantic segmentation datasets, where it outperformed previous work by large margins in all evaluations.

## 2 RELATED WORK

### 2.1 DOMAIN GENERALIZABLE SEMANTIC SEGMENTATION

The objective of domain generalization is to develop models that generalize well to *unseen* domains (Muandet et al., 2013; Li et al., 2018b). Early methods (Li et al., 2017; 2018a; Pan et al., 2018; Nam et al., 2021; Zhou et al., 2021) primarily focused on classification tasks. Recently, significant progress has been made in semantic segmentation (Yue et al., 2019b; Lee et al., 2022; Zhao et al.,

2022; Chattopadhyay et al., 2023; Kim et al., 2023a). Yue et al. (2019b) suggest learning features invariant to random style variations in the input. Methods like Lee et al. (2022); Zhao et al. (2022) simulate diverse style spaces by manipulating the channel-wise means and standard deviations of features. Additionally, Chattopadhyay et al. (2023) introduced a frequency-domain randomization technique, particularly for strong augmentation in high-frequency regions. Furthermore, Kim et al. (2023a) collected a variety of images with different styles from web repositories to enhance generalization performance. Also, some recent work explores alternative model architectures for the purpose. For instance, Ding et al. (2023) propose HGFormer, a hierarchical grouping transformer that integrates both local and global feature interactions to improve generalization in unseen domains. Luo et al. (2024) demonstrate that network pruning can enhance domain generalization by reducing model complexity and increasing robustness. DAFormer (Hoyer et al., 2022) introduces domain-adaptive semantic segmentation by incorporating architectural refinements and training strategies that enhance robustness across diverse domains. Meanwhile, DGInStyle (Jia et al., 2024) employs image diffusion models to generate diverse stylized versions of training images, simulating various domain shifts; this approach enables models to learn domain-invariant features, effectively improving their robustness to diverse input distributions during inference.

However, the listed methods still adopt joint finetuning, despite its negative impact on generalization ability, which remains a common practice in existing frameworks. To address this generalization issue in previous work, we propose a novel decoupled finetuning strategy called DeFT, and present a dedicated training algorithm based on EMA.

## 2.2 Weight averaging for model ensembling

Weight averaging has been explored as an effective method for leveraging the historical training trajectories of deep neural networks to improve generalization performance. Snapshot ensembling (SSE) (Huang et al., 2017) and fast geometric ensembling (FGE) (Garipov et al., 2018) were early attempts to utilize weight trajectories from historical training by employing cyclic learning rates to guide the learning process through multiple local minima, which are then saved as ensemble members. Building on FGE, stochastic weight averaging (SWA) (Izmailov et al., 2018) updates a pretrained model using a cyclical or high constant learning rate, gathers model parameters during training, and averages them to form a model ensemble. Extending from SWA, trainable weight averaging (TWA) (Li et al., 2023) introduced a technique that allows for weight averaging with adjustable coefficients. Additionally, model soups (Wortsman et al., 2022a) demonstrated that averaging the weights of multiple models finetuned with different hyperparameters can improve both accuracy and robustness.

We adopt a variant of EMA as temporal ensembling in the model parameter space for DeFT, allowing the model to learn task-relevant knowledge while minimizing overfitting to the source domain.

## 2.3 Robust finetuning for out-of-domain generalization

Robust finetuning using pretrained weights enhances out-of-domain (OOD) performance. Recent studies (Wen et al., 2021; Gouk et al., 2021) demonstrate that leveraging pretrained models can significantly booster robustness on OOD datasets. Moreover, the finetuning process plays a critical role in improving OOD generalization capability. Research (Nagarajan & Kolter, 2019; Lin & Zhang, 2019; Gouk et al., 2021; Li & Zhang, 2021) indicates that generalization performance is affected by the distance between the initial and finetuned models: as this distance increases, generalization tends to decline. WiSE-FT (Wortsman et al., 2022b) shows significant improvements in OOD generalization by linearly interpolating pretrained weights with finetuned ones during inference. LP-FT (Kumar et al., 2022) demonstrates that simultaneously finetuning both the final linear layer and the feature backbone can distort pretrained features, and proposes a two-stage training strategy: first, warming up the decoder while freezing the encoder, then finetuning the entire network. Lastly, Tian et al. (2023) introduces per-layer regularization, which automatically learns constraints for more accurate finetuning.

Our method adopts a two-step training strategy motivated by LP-FT, but it is clearly different from LP-FT: after warming up the decoder, we finetune encoder and decoder in a disjoint manner with the proposed DeFT framework, rather than jointly finetuning them as in LP-FT.

## 3 METHOD

We consider training a domain-generalizable segmentation model, $f = d \circ e$, with an encoder $e$ and decoder $d$ using labeled images from a single source domain, where the image height, width, and the number of semantic classes are denoted by $h$, $w$, and $c$, respectively. Our framework, dubbed DeFT, consists of two stages: warming up the decoder while freezing the pretrained encoder (Section 3.1), followed by decoupled finetuning of the encoder and decoder (Section 3.2).

### 3.1 WARMING UP THE DECODER WITH A PRETRAINED ENCODER

The first step of DeFT is to warm up the decoder on the source domain dataset using the pretrained encoder that remains frozen. For an input image $X$ and its ground truth $Y$, let $P = f(X) = d(e(X))$ be the class probability map, where $P \in \mathbb{R}^{h \times w \times c}$. Let $\mathcal{L}_{\text{ce}}(P, Y)$ denote a standard pixel-wise cross-entropy loss, which is given by:

$$\mathcal{L}_{\text{ce}}(P, Y) = -\frac{1}{h \cdot w} \sum_{i=1}^{h \cdot w} Y_i^\top \cdot \log(P_i), \tag{1}$$

where $i$ is the pixel index and $Y_i$ is the one-hot vector of the ground truth for pixel $i$. Then, the weights of the randomly initialized decoder $d$ are updated using gradients of the cross-entropy loss $\mathcal{L}_{\text{ce}}(P, Y)$, while the pretrained encoder is frozen. This warming up stage enables the decoder to learn the target task without distorting the generalizable knowledge of the pretrained encoder.

### 3.2 DECOUPLED FINETUNING OF ENCODER AND DECODER

After warming up the decoder, the encoder and decoder are fine-tuned in a *decoupled manner*. During decoupled finetuning, they are assigned as one of two components, the retentive component (RC) and the adaptive component (AC), but are different from each other. Then AC is updated using gradients of the training loss from the source domain, while RC is updated by an exponential moving average (EMA) scheme to retain its generalization capability.

To decouple the encoder and decoder then finetune both, we define two distinct pathways, which can be represented as two segmentation models $f_1$ and $f_2$ that share the same architecture but have opposing configurations for AC and RC. Let $f_1 = \tilde{d}_1 \circ e_1$ and $f_2 = d_2 \circ \tilde{e}_2$ be the segmentation models, where $\tilde{d}_1$ and $\tilde{e}_2$ are the RCs, and $d_2$ and $e_1$ are the ACs. The weights of ACs, $d_2$ and $e_1$, are updated using the cross-entropy loss $\mathcal{L}_{\text{ce}}(P^{(1)}, Y)$ and $\mathcal{L}_{\text{ce}}(P^{(2)}, Y)$, where $P^{(1)} = f_1(X)$ and $P^{(2)} = f_2(X)$ are the predictions of $f_1$ and $f_2$, respectively. On the other hand, RC in one model is not updated by the cross-entropy loss but by the exponential moving average of AC in the other, with an update ratio $\beta$:

$$\tilde{\theta}_{d_1}^{t+1} = \beta\tilde{\theta}_{d_1}^t + (1-\beta)\theta_{d_2}^t, \quad \tilde{\theta}_{e_2}^{t+1} = \beta\tilde{\theta}_{e_2}^t + (1-\beta)\theta_{e_1}^t, \tag{2}$$

where $\tilde{\theta}_{d_1}^t$ and $\tilde{\theta}_{e_2}^t$ are the weights of $\tilde{d}_1$ and $\tilde{e}_2$ at the $t$-th iteration, respectively. Updates of ACs and RCs are conducted simultaneously during the second stage.

### 3.3 OUR FINAL MODEL FOR INFERENCE

DeFT produces two encoders, $e_1$ and $\tilde{e}_2$, and two decoders, $\tilde{d}1$ and $d_2$. As the two RCs better preserve their initial states which are more generalizable and thus have superior generalization capability compared to the ACs, we set our final model as the combination of the two RCs from $f_1$ and $f_2$: $f_{\text{final}} = \tilde{d}_1 \circ \tilde{e}_2$. Since each RC from different pathways is updated based on its counterpart AC in Eq. (2), their feature distributions are implicitly aligned, properly adapting to each other.

### 3.4 EMPIRICAL JUSTIFICATION FOR DEFT

The decoupled finetuning improves the model's generalization capability by enabling the encoder and decoder to be trained independently, each benefiting from less-overfitted decision boundaries or features derived from their respective generalized counterparts. Moreover, it can tighten the

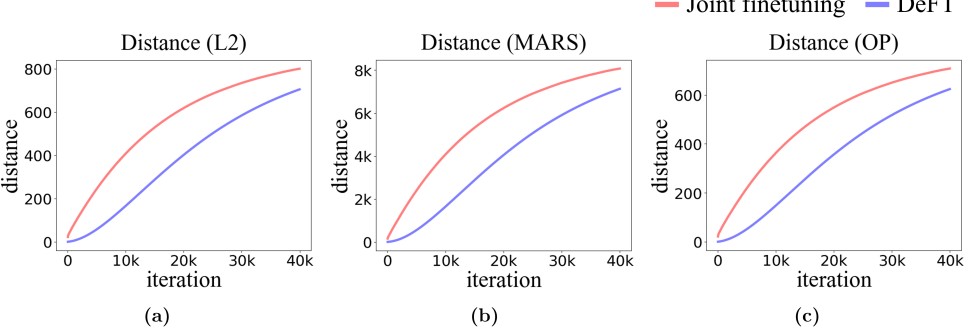

Figure 3: We empirically demonstrate that the model jointly finetuned tends to move further away from its initial parameters than the model finetuned with DeFT. We measured the distance between the current and initial parameters of each model using three metrics: (a) L2 norm (Nagarajan & Kolter, 2019), (b) MARS norm (Gouk et al., 2021), and (c) operator norm (Long & Sedghi, 2020). The final model $f_{\text{final}} = \tilde{d}_1 \circ \tilde{e}_2$ was used to measure the distance for DeFT. The results show that the model finetuned with DeFT exhibits a shorter distance from its initial parameters than the jointly finetuned one, resulting in better generalization performance.

generalization bounds of the model, as it reduces both *the number of parameters* to be optimized at each pathway and *the distance from initial parameters*, *i.e.*, the distance between the learned parameters and their initial values.

DeFT divides a single optimization objective of joint finetuning, which handles all the parameters in the model simultaneously, into two separate optimization objectives: one for the encoder and one for the decoder. As a result, each of the networks $f_1$ and $f_2$ is trained on a separate objective with fewer parameters to optimize compared to the original model. This leads to a tighter generalization bound for each module than the joint finetuning (Du et al., 2018b; Long & Sedghi, 2020).

We also empirically demonstrate that DeFT reduces the distance from initial parameters, which is one of the critical factors for the model's generalization bounds (Nagarajan & Kolter, 2019; Long & Sedghi, 2020; Gouk et al., 2021; Li & Zhang, 2021). Thanks to the EMA update scheme in DeFT, RCs are updated to maintain their initial states, partially retaining their initial weights. RCs, $\tilde{e}_2$ and $\tilde{d}_1$, which comprise our final model, exhibit a shorter distance from their initialization as shown in Figure 3, resulting in better domain generalization capability.

Moreover, we compare DeFT to other methods which explicitly regularize the distance from initial parameters: using low learning rates and adding a distance regularization into optimization objective. The results in Table 2 show that simply reducing the learning rate does not always lead to performance improvement, rather degrading its generalization performance due to the increased the risk of falling in local minima. For the distance regularization, we add a regularization term to the optimization objective to regularize the sum of the Euclidean distances from initial parameters. Let $L$ be the number of layers in the model, and, $W_i$ and $W_i^{(0)}$ be weights of the model and its initial parameters of the $l$-th layer. Then the revised training loss after warming up the decoder is given by:

$$\mathcal{L}_{\text{training}} = \mathcal{L}_{\text{ce}} + \alpha \cdot \sum_{i=1}^{L} \|W_i - W_i^{(0)}\|, \tag{3}$$

where $\|\cdot\|$ denotes the Frobenius norm for matrices and the Euclidean norm for vectors, respectively. The results reported in Table 3 demonstrate that DeFT outperforms joint finetuning, even the distance regularization is applied. The result also suggests that distance regularization can be applied orthogonally to DeFT. In addition to the empirical justification, we provide theoretical foundation of DeFT in Section B.

## 4 EXPERIMENTS

In this section, we first describe the experimental settings and implementation details, followed by a demonstration of the effectiveness of our DeFT framework through a series of extensive experiments, including various ablation studies.

Table 2: Analysis of the impact of learning rate. We investigate whether reducing the learning rate after the warm-up stage can lead to better generalization. All the methods were trained on GTAV using ResNet-50 backbone. The results demonstrate that a small learning rate increases the risk of falling in local minima, resulting in suboptimal performance.

| Method | Learning rate | Cityscapes | BDD100K | Mapillary | Avg. |
|---|---|---|---|---|---|
| Joint Finetuning | 1e-2 | 42.32 | 40.33 | 44.88 | 42.51 |
| | 3e-3 | 44.87 | 42.10 | 49.38 | 45.45 |
| | 1e-3 | 44.01 | 39.47 | 47.45 | 43.64 |
| | 1e-4 | 40.84 | 37.38 | 44.83 | 41.02 |
| DeFT | 1e-2 | **50.06** | **43.17** | **50.51** | **47.91** |

Table 3: Impact of distance-based regularization. All the methods were trained on GTAV using ResNet-50 backbone. DeFT outperformed joint finetuning in both settings. The result also suggests that the distance regularization is orthogonal to DeFT and further improve its performance.

| Method | Cityscapes | BDD100K | Mapillary | Avg. |
|---|---|---|---|---|
| Joint Finetuning | 42.32 | 40.33 | 44.88 | 42.51 |
| Joint Finetuning + Distance Regularization | 46.06 | 40.80 | 47.93 | 44.93 |
| DeFT | 50.06 | 43.17 | 50.51 | 47.91 |
| DeFT + Distance Regularization | **51.09** | **43.46** | **51.58** | **48.71** |

## 4.1 EXPERIMENTAL SETUP

**Datasets.** We used three real-world datasets, Cityscapes (Cordts et al., 2016), BDD-100K (Yu et al., 2020), and Mapillary (Neuhold et al., 2017), and two synthetic datasets, GTAV (Richter et al., 2016) and SYNTHIA (Ros et al., 2016) for the experiment. Cityscapes is a real-world urban driving scene dataset, comprising 2,985 images for training and 500 for validation. BDD-100K is another real-world urban driving scene dataset, and we used the 1,000 validation images for evaluation. Mapillary consists of 25,000 images collected from various worldwide locations, and we used 2,000 validation images for evaluation. GTAV contains 24,966 images generated from the Grand Theft Auto V (GTAV) game engine, split into 12,403 images for training and 6,382 for validation. SYNTHIA is a photo-realistic synthetic urban scene dataset, consisting of 9,400 images. We used 6,382 validation images for evaluation.

**Network architecture.** We utilized DeepLab v3+ (Chen et al., 2018) as the segmentation model with ImageNet (Deng et al., 2009) pretrained ResNet-(50/101) (He et al., 2016) backbone networks. During training, we introduced two segmentation models that share the same architecture.

**Implementation details.** The model was trained with a batch size of 4 through SGD with a momentum of 0.9. For the warm-up stage, the model was trained for 2K iterations for Cityscapes and 8K iterations for GTAV, with a learning rate of 1e-2 and a weight decay of 5e-3. During the decoupled finetuning, the model was trained for 40K iterations with a learning rate of 1e-2 and a weight decay of 5e-4. We employed a polynomial learning rate decay schedule with a power of 0.9. For data augmentation, we adopted color jittering, Gaussian blurring, random horizontal flipping with a probability of 0.5, random scaling in the range [0.5, 2.0], and random cropping with a size of $768 \times 768$. The weight update ratio $\beta$ was set to 0.9999. We used the mean Intersection-over-Union (mIoU) as the evaluation metric. We excluded the auxiliary cross-entropy loss applied to the encoder, which has been widely adopted in previous work (Zhao et al., 2017; Pan et al., 2018; Lee et al., 2022; Zhao et al., 2022; Chattopadhyay et al., 2023; Ahn et al., 2024), as it degrades OOD generalization capability.

## 4.2 COMPARISON WITH STATE OF THE ART

We conducted a series of experiments to evaluate the effectiveness of DeFT. DeFT was compared with existing domain generalization methods, including IBN-Net (Pan et al., 2018), DRPC (Yue

Table 4: Quantitative result comparison in mIoU (%) using ResNet-50 and ResNet-101 backbones. The model was trained on GTAV and evaluated on Cityscapes (C), BDD100K (B), and Mapillary (M).

| Methods | ResNet-50 | | | | ResNet-101 | | | |
|---|---|---|---|---|---|---|---|---|
| | C | B | M | Avg. | C | B | M | Avg. |
| Baseline | 35.16 | 29.71 | 31.29 | 32.05 | 35.73 | 34.06 | 33.42 | 34.40 |
| IBN-Net (Pan et al., 2018) | 33.85 | 32.30 | 37.75 | 34.63 | 37.37 | 34.21 | 36.81 | 36.13 |
| DRPC (Yue et al., 2019a) | 37.42 | 32.14 | 34.12 | 34.56 | 42.53 | 38.72 | 38.05 | 39.77 |
| ISW (Choi et al., 2021) | 36.58 | 35.20 | 40.33 | 37.37 | 37.20 | 33.36 | 35.57 | 35.38 |
| WildNet (Lee et al., 2022) | 44.62 | 38.42 | 46.09 | 43.04 | 45.79 | 41.73 | 47.08 | 44.87 |
| SAN-SAW (Peng et al., 2022) | 39.75 | 37.34 | 41.86 | 39.65 | 45.33 | 41.18 | 40.77 | 42.43 |
| DIRL (Xu et al., 2022) | 41.04 | 39.15 | 41.60 | 40.60 | - | - | - | - |
| SHADE (Zhao et al., 2022) | 44.65 | 39.28 | 43.34 | 42.42 | 46.66 | 43.66 | 45.50 | 45.27 |
| PASTA (Chattopadhyay et al., 2023) | 44.12 | 40.19 | _47.11_ | 43.81 | 45.33 | 42.32 | 48.60 | 45.42 |
| TLDR (Kim et al., 2023b) | _46.51_ | _42.58_ | 46.18 | _45.09_ | _47.58_ | _44.88_ | _48.80_ | _47.09_ |
| BlindNet (Ahn et al., 2024) | 45.72 | 41.32 | 47.08 | 44.71 | - | - | - | - |
| DeFT (Ours) | **50.06** | **43.17** | **50.51** | **47.91** | **52.14** | **45.16** | **53.15** | **50.15** |

Table 5: Quantitative result comparison in mIoU (%) using ResNet-50 backbone. The model was trained on Cityscapes and evaluated on BDD-100K (B), SYNTHIA (S), and GTAV (G).

| Methods | B | S | G | Avg. |
|---|---|---|---|---|
| Baseline | 44.96 | 23.29 | 42.55 | 36.93 |
| IBN-Net (Pan et al., 2018) | 48.56 | 26.14 | 45.06 | 39.92 |
| DRPC (Yue et al., 2019a) | 49.86 | 26.58 | 45.62 | 40.69 |
| ISW (Choi et al., 2021) | 50.74 | 26.20 | 45.00 | 40.64 |
| WildNet (Lee et al., 2022) | 50.94 | 27.95 | 47.01 | 41.97 |
| SAN-SAW (Peng et al., 2022) | _52.95_ | 28.32 | 47.28 | _42.85_ |
| DIRL (Xu et al., 2022) | 51.80 | 26.50 | 46.52 | 41.61 |
| SHADE (Zhao et al., 2022) | 50.95 | 27.62 | _48.61_ | 42.39 |
| BlindNet (Ahn et al., 2024) | 51.84 | _28.51_ | 47.97 | 42.77 |
| DeFT (Ours) | **53.12** | **28.87** | **48.72** | **43.57** |

et al., 2019a), ISW (Choi et al., 2021), WildNet Lee et al. (2022), SAN-SAW (Peng et al., 2022), DIRL Xu et al. (2022), SHADE Zhao et al. (2022), PASTA (Chattopadhyay et al., 2023), TLDR (Kim et al., 2023b), and BlindNet (Ahn et al., 2024), using five datasets—(C)ityscapes, (B)DD-100K, (M)apillary, (S)YNTHIA, and (G)TAV, and two different backbone networks—ResNet-50 and ResNet-101. To evaluate the generalization ability of our method on various unseen domains, we conducted experiments in two scenarios: 1) the model was trained on GTAV and evaluated on Cityscapes, BDD-100K, and Mapillary, or 2) the model was trained on Cityscapes and evaluated on BDD-100K, SYNTHIA, and GTAV. For the first case, the results in Table 4 show that our method outperforms all other methods by a large margin when trained on GTAV, using either ResNet-50 or ResNet-101 as a backbone. Similarly, the results in Table 5 demonstrate that our method also outperforms all other methods in the second case, where the model was trained on Cityscapes with ResNet-50 backbone.

## 4.3 ABLATION STUDIES

In this subsection, we study the individual contribution and effectiveness of each component within our method. For the all experiments, the model was trained on GTAV and evaluted on Cityscapes, BDD100K and Mapillary with ResNet-50 backbone.

**Ablation study of the impact of individual component for training.** To investigate the contribution of individual component during training, we investigated the impact of various training components and measured its performance. We considered four different components for the ex-

Table 6: Ablation study of the impact of individual component for training. Aux. and Aug. denote the auxiliary cross-entropy loss attached to the encoder and data augmentation, respectively. All the methods were trained on GTAV using ResNet-50 backbone. Warm-Up represent warming up the decoder while freezing the encoder before finetuning, and DeFT is the proposed decoupled finetuning.

| w/o Aux. | Aug. | Warm-Up | DeFT | Cityscapes | BDD100K | Mapillary | Avg. |
|---|---|---|---|---|---|---|---|
| | | | | 35.16 | 29.71 | 31.29 | 32.05 |
| ✓ | | | | 36.58 | 34.49 | 39.08 | 36.72 |
| ✓ | ✓ | | | 40.77 | 37.87 | 43.39 | 40.66 |
| ✓ | ✓ | ✓ | | 42.32 | 40.33 | 44.88 | 42.51 |
| ✓ | ✓ | ✓ | ✓ | **50.06** | **43.17** | **50.51** | **47.91** |

Table 7: Ablation study of the impact of the decoupled finetuning strategy.

| Finetuning strategy | Cityscapes | BDD100K | Mapillary | Avg. |
|---|---|---|---|---|
| Joint finetuning | 42.32 | 40.33 | 44.88 | 42.51 |
| Joint finetuning + EMA | 48.30 | 42.29 | 49.02 | 46.54 |
| DeFT | **50.06** | **43.17** | **50.51** | **47.91** |

Table 8: Analysis on the impact of final model configuration.

| ID | $e_1$ (AC) | $\tilde{e}_2$ (RC) | $\tilde{d}_1$ (RC) | $d_2$ (AC) | Cityscapes | BDD100K | Mapillary | Avg. |
|---|---|---|---|---|---|---|---|---|
| I | ✓ | | | ✓ | 39.30 | 37.41 | 43.14 | 39.95 |
| II | ✓ | | ✓ | | 43.15 | 39.82 | 45.55 | 42.84 |
| III | | ✓ | | ✓ | 47.29 | 41.84 | 49.33 | 46.15 |
| IV | | ✓ | ✓ | | **50.06** | **43.17** | **50.51** | **47.91** |

periments: removing the auxiliary cross-entropy loss attached to the encoder (w/o Aux.), data augmentation (Aug.), decoder warming up (Warm-up) and our DeFT framework (DeFT). Note that all the settings except the last row (DeFT) conducted joint finetuning, instead of decoupled finetuning. The results in Table 6 show that each component contributes to the performance, and applying all of them improves the most.

**Ablation study on the decoupled finetuning strategy.** we conducted an experiment to investigate the effect of decoupled finetuning and that of weight ensemble separately. To be specific, we jointly finetuned the encoder and decoder, and considered their EMA versions as the final model for evaluation. The EMA update ratio $\beta$ was set to 0.9999, the same as DeFT. The results in the Table 7 show that the proposed decoupled finetuning strategy better preserves generalizable knowledge of the pretrained encoder and decoder than joint finetuning.

**Ablation study on final model configuration.** We set our final model as the combination of two RCs, *i.e.*, EMA encoder $\tilde{e}_2$ and EMA decoder $\tilde{d}_1$, as RCs preserve the rich and generalizable knowledge. To investigate this, we conducted additional ablation study on various combination of RCs and ACs. We measured the performance of each combination when the whole training ended, and the results are listed in Table 8. The experiments show that superior generalization capability of RC (Exp. II, III and IV) than AC (Exp. I), where using both of RCs outperformed all other settings by a large margin.

**Ablation study on the impact of the update ratio $\beta$.** We employed the exponential moving average as an update scheme for RC with the update ratio $\beta$ in Eq. (2). To investigate the impact of $\beta$, we conducted additional experiments by varying the values of $\beta$. The results in Table 9 demonstrate that assigning a higher weight to the model's initial parameters yields better generalization performance.

Table 9: Ablation study of the impact of the update ratio $\beta$ in Eq. (2).

| EMA update ratio ($\beta$) | Cityscapes | BDD100K | Mapillary | Avg. |
|---|---|---|---|---|
| 0.99 | 44.37 | 40.79 | 46.81 | 43.99 |
| 0.999 | 46.19 | 42.14 | 48.81 | 45.71 |
| 0.9999 | **50.06** | **43.17** | **50.51** | **47.91** |

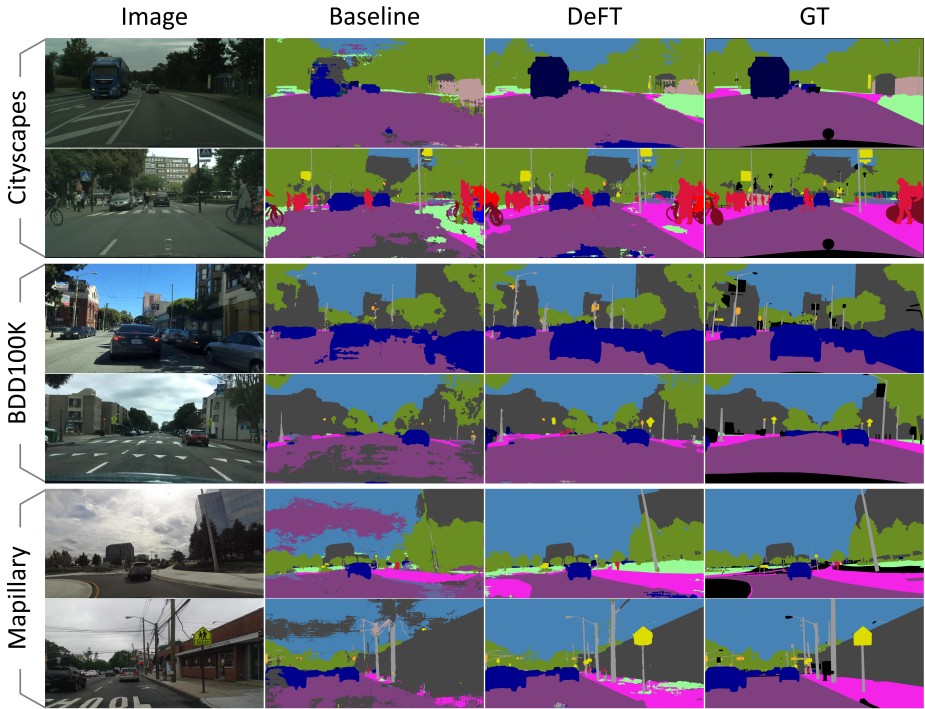

Figure 4: Qualitative results of DeFT and its baseline. The model was trained on the GTAV dataset using ResNet-50 and tested on the Cityscapes, BDD100K, and Mapillary datasets.

## 5 CONCLUSION

In this paper, we have demonstrated the detrimental effects of jointly finetuning the encoder and decoder in semantic segmentation models on domain generalization. Our empirical analysis revealed that this common practice leads to overfitting on the source domain, thereby degrading the model's generalization capability. To address this issue, we introduced DeFT, a novel and effective training framework that decouples the finetuning of the encoder and decoder. This decoupled finetuning prevents them from being trained based on their counterpart, which might be overfitted to the source domain, resulting in improved generalization capability. DeFT operates in two stages. In the first stage, we warm up the decoder while keeping the pretrained encoder frozen. In the second stage, we decouple the finetuning process by employing two parallel encoder-decoder pathways, each consisting of adaptive components (ACs) and retentive components (RCs). The ACs are updated through standard backpropagation on the source domain, while the RCs are updated using an exponential moving average of the ACs' parameters. This approach enables the RCs to learn task-relevant information while maintaining their generalization ability. Our extensive experiments demonstrate that DeFT consistently outperforms existing methods in domain generalizable semantic segmentation.

**Limitation and discussion.** Although DeFT has demonstrated impressive performance improvement through a large number of experiments, there is still room for further theoretical analysis. Also, there might be better design choices for RCs, although the EMA models showed their generalization capability. We believe that building a more concrete theoretical foundation and exploring a better alternative configurations for the RCs will be promising future research directions.

ACKNOWLEDGEMENTS AND DISCLOSURE OF FUNDING

This work was supported by Samsung Research Funding & Incubation Center of Samsung Electronics under Project Number SRFC-IT1801-52 and the Institute of Information & Communications Technology Planning & Evaluation (IITP) with a grant funded by the Ministry of Science and ICT (MSIT) of the Republic of Korea in connection with the Global AI Frontier Lab International Collaborative Research. (No. RS-2024-00469482 & RS-2024-00509258)

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

## A    ALGORITHMS FOR EACH STAGE OF DEFT

In this sections, we present PyTorch-like pseudocodes for each stage of DeFT. Algorithm 1 describes the training procedure for decoder warm-up, and Algorithm 2 describes the training procedure for decoupled finetuning and the configuration of the final model for inference.

---

**Algorithm 1** Pseudocode for Decoder Warm-Up, PyTorch-like

```
1  # e: pretrained encoder e
2  # d: randomly initialized decoder d
3  # CE: standard pixel-wise cross entropy loss 𝓛ce
4
5  freezing_weights(e)
6
7  for x, y in source_loader:        # load a minibatch x with n samples
8      p = d(e(x))                   # prediction P = f(X) = d(e(X))
9      L = CE(p, y)                  # calculate loss
10
11     L.backward()
12     update(d)                     # only the decoder d is updated
```

---

**Algorithm 2** Pseudocode for Decoupled Finetuning and Inference, PyTorch-like

```
1  # e: warmed-up encoder e, the same as the pretrained encoder
2  # d: warmed-up decoder d
3  # CE: standard pixel-wise cross entropy loss 𝓛ce
4  # beta: EMA update ratio β
5
6  e1 = copy(e)                      # AC encoder e₁
7  e2 = copy(e)                      # RC encoder ẽ₂
8
9  d1 = copy(d)                      # RC decoder d̃₁
10 d2 = copy(d)                      # AC decoder d₂
11
12 freezing_weights(e2)
13 freezing_weights(d1)
14
15 for x, y in source_loader:        # load a minibatch x with n samples
16     p1 = d1(e1(x))                # prediction P⁽¹⁾ from f₁ = d̃₁ ∘ e₁
17     L1 = CE(p1, y)                # calculate loss
18
19     p2 = d2(e2(x))                # prediction P⁽²⁾ from f₂ = d₂ ∘ ẽ₂
20     L2 = CE(p2, y)                # calculate loss
21
22     L1.backward()
23     L2.backward()
24
25     update(e1)                    # update AC encoder e₁
26     update(d2)                    # update AC decoder d₂
27
28     update_EMA(e2, e1, beta)      # update RC encoder ẽ₂
29     update_EMA(d1, d2, beta)      # update RC decoder d̃₁
30
31 def update_EMA(rc, ac, beta):
32     for param_rc, param_ac in zip(rc.parameters(), ac.parameters()):
33         param_rc = beta * param_rc + (1 - beta) * param_ac
34
35 def inference(x):
36     return d1(e2(x))              # final model f_final = d̃₁ ∘ ẽ₂
```

# B  THEORETICAL FOUNDATION OF DeFT

This section theoretically justifies the motivation and design choices of DeFT to validate its efficacy. To be specific, we discuss (1) the issues of joint finetuning, and (2) how DeFT resolves them.

Kumar et al. (2022) showed that jointly finetuning a well-generalizable encoder with a randomly initialized decoder distorts the representation of the encoder, and proved this theoretically for the case of a two-layer linear neural network. Let us suppose finetuning a two-layer linear network $f_{B,v}$ consisting of an encoder $B \in \mathbb{R}^{k \times d}$ and a decoder head $v \in \mathbb{R}^k$. For in-distribution (ID) data $X \in \mathbb{R}^{n \times d}$ and its corresponding label $y \in \mathbb{R}^n$, the loss to be optimized is $L(\hat{y}, y) = ||\hat{y} - y||_2^2$, where $\hat{y} = f_{B,v}(x) = XB^\top v$ is the prediction of the network $f$ for $X$. Then, the gradient of $B$ for the loss $L$ is as follows:

$$\nabla_B L(\hat{y}, y) = 2v(XB^\top v - y)^\top X.$$

Considering an out-of-distribution (OOD) data point $u \in \mathbb{R}^d$ which is orthogonal to the ID subspace $S = \text{rowspace}(X)$. Then the feature representation of the updated encoder for $u$, denoted by $B'u$, where $B' = B - \lambda \nabla_B L(\hat{y}, y)$, is as follows:

$$B'u = (B - \lambda \nabla_B L(\hat{y}, y))u = Bu - \lambda \nabla_B L(\hat{y}, y)u = Bu - 2v(XB^\top v - y)^\top Xu.$$

Note that $u$ is orthogonal to the rowspace of $X$, and thus $2v(XB^\top v - y)^\top Xu$ goes to 0. Therefore, in such case, changes in $B$ with respect to the ID data $X$ cannot affect the feature representation of the OOD data $u$. However, the decoder head $v$ also changes since loss gradients for updating the encoder and decoder are coupled, which is referred to as "balancedness" in Du et al. (2018a). Then the change of the decoder head affects the predictions for OOD data $U$, which is given by

$$UB'^\top v' = UB'^\top (v - \lambda_v \nabla_v L) = UB'^\top v - \lambda_v UB'^\top \nabla_v L$$

Since the representations for $U$ are same during the update, i.e. $UB = UB'$, the predictions for $U$ can be distorted as $\lambda_v UB'^\top \nabla_v L$, where $\nabla_v L = 2BX^\top(XB^\top v - y)$. Such distortion in prediction can impair the generalization capability on OOD data as $\nabla_v L$ is calculated solely based on ID data $X$. As a result, the entire model becomes less capable of handling OOD data by the joint finetuning. This suggests the need to decouple one module from being affected by the distortion of another module.

On the other hand, consider a linear probing case where the encoder is fixed and no feature distortion occurs. According to Lemma A.14 in Kumar et al. (2022), the upper bound of OOD error of the linear probing is:

$$\sqrt{L_{\text{ood}}(v_{\text{lp}}^\infty, B_0)} \leq \left(\frac{c_\delta}{\cos \theta_{\max}(S, R)}\right)^2 d(B_0, B_\star)\|w_\star\|_2,$$

where $L_{\text{ood}}(v_{\text{lp}}^\infty, B_0)$ is OOD error with the frozen initial encoder $B_0$ and linearly probed decoder head $v_{\text{lp}}^\infty$, $\cos \theta_{\max}(S, R)$ is the cosine of the largest angle between $S = \text{rowspace}(X)$ and $R = \text{rowspace}(B_0)$, $w_\star = B_\star v_\star$ with the optimal encoder $B_\star$ and decoder $v_\star$, and $d(B_0, B_\star)$ is the distance between $B_0$ and $B_\star$. A rigorous proof of the above upper bound can be found in Kumar et al. (2022). This demonstrates that the upper bound of OOD error of the linear probing is inversely proportional to the difference between the pretrained encoder and the "optimal" encoder, which shows the lowest errors for both ID and OOD data. In other words, in the linear probing case, a decoder coupled with more generalizable encoder results in a more generalizable final model.

Based on this, we first decouple the encoder and decoder to prevent each module from being distorted by their jointly finetuned counterparts, coupling them with counterparts which are not affected by distribution (i.e., domain) shift thus are well generalizable. Through the empirical analysis of the distance from initialization in Figure 3, it can be inferred that DeFT suppresses such "distortion" from the initial states during training, resulting in better generalization capability.

The optimization behavior from the perspective of individual network modules, such as encoders and decoders, as well as the interactions between these modules during the optimization process, seems to remain relatively underexplored. We expect that further foundational study in this direction might pave the way for more rigorous theoretical analysis of DeFT.

## C  VERSATILITY OF DeFT AT A TRANSFORMER BACKBONE

We evaluated DeFT using the MiT-B5 (Xie et al., 2021) transformer backbone. All models were trained on GTAV and evaluated on Cityscapes, BDD100K, and Mapillary. As shown in Table 10, DeFT outperformed existing methods, such as DAFormer (Hoyer et al., 2022) and the combination of DAFormer and DGinStyle (Jia et al., 2024), using the same backbone. Notably, DeFT achieved this outstanding performance without additional modifications, unlike DAFormer, which adapts its model architecture for domain generalization, or DGinStyle, which relies heavily on extreme data augmentation. We believe that incorporating such strategies could further enhance DeFT's performance. These results demonstrate that DeFT is a versatile training strategy applicable across various backbones.

Table 10:  Comparison between DeFT and other methods incorporating MiT-B5 backbone.

| Segformer / MiT-B5 | Cityscapes | BDD100K | Mapillary | Avg. |
|---|---|---|---|---|
| DAFormer (Hoyer et al., 2022) | 52.65 | 47.89 | 54.66 | 51.73 |
| DAFormer + DGInStyle (Jia et al., 2024) | 55.31 | **50.82** | 56.62 | 54.25 |
| **DeFT (Ours)** | **57.16** | 49.32 | **59.99** | **55.49** |

## D  OTHER UPDATE SCHEMES FOR RETENTIVE COMPONENTS (RCs)

We conducted additional experiments by replacing the update scheme for RC in DeFT (i.e., EMA) with two different weight ensemble methods.

(A) Exponentially decreasing the later ensemble coefficient:

$$\theta_t^{\text{RC}} = \frac{\alpha * \theta_{t-1}^{\text{RC}} + \theta_t^{\text{AC}}}{T}, \quad T = \sum_{i=0}^{t-1} \alpha^i, \quad \alpha > 1. \tag{4}$$

(B) Simply averaging the latest AC weights with the initial weights, similar to WiSE-FT (Wortsman et al., 2022b):

$$\theta_t^{\text{RC}} = 0.5 * \theta_t^{\text{AC}} + 0.5 * \theta_0. \tag{5}$$

All models were trained on GTAV and evaluated on Cityscapes, BDD100K, and Mapillary. As shown in Table 11, the EMA biased towards initialization employed in DeFT clearly outperformed the other weight ensemble methods.

Table 11: Comparison between the EMA and other RC update schemes.

| Update scheme | Cityscapes | BDD100K | Mapillary | Avg. |
|---|---|---|---|---|
| (A) Exponentially decreasing | 44.46 | 40.86 | 46.72 | 44.01 |
| (B) WiSE-FT | 45.03 | 40.47 | 47.04 | 44.18 |
| (C) **EMA (DeFT)** | **50.06** | **43.17** | **50.51** | **47.91** |

## E  ADDITIONAL QUALITATIVE RESULTS

In this section, we provide additional qualitative results on Cityscapes (Cordts et al., 2016) in Figure 5, BDD100K (Yu et al., 2020) in Figure 6, and Mapillary (Neuhold et al., 2017) in Figure 7, respectively, which are not presented in the main sections due to the space limit. We used the model trained on the GTAV (Richter et al., 2016) using ResNet-50 (He et al., 2016) backbone. We also provide the color code of 19 classes in Figure 8.

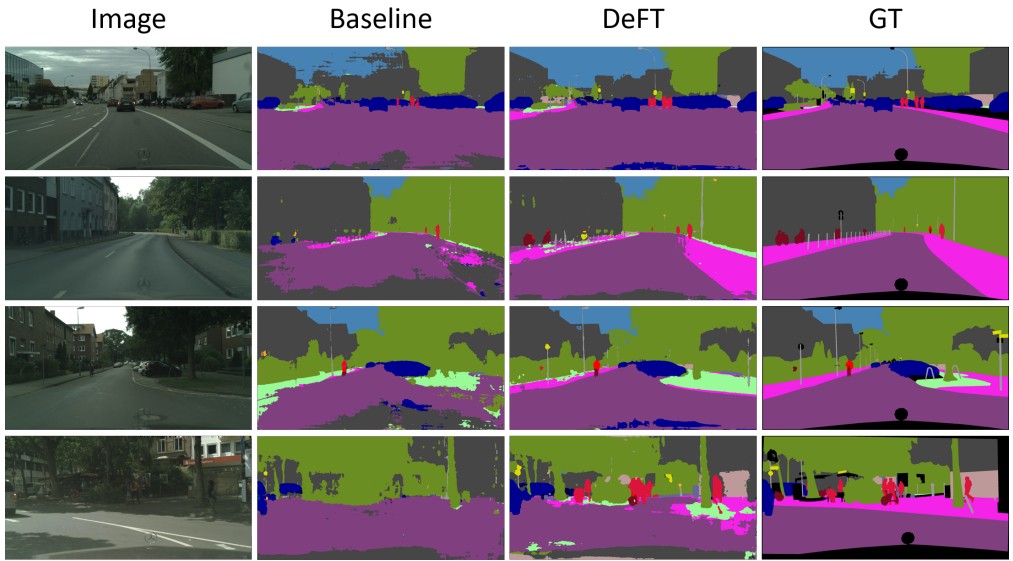

Figure 5: Qualitative results of DeFT and its baseline on Cityscapes dataset.

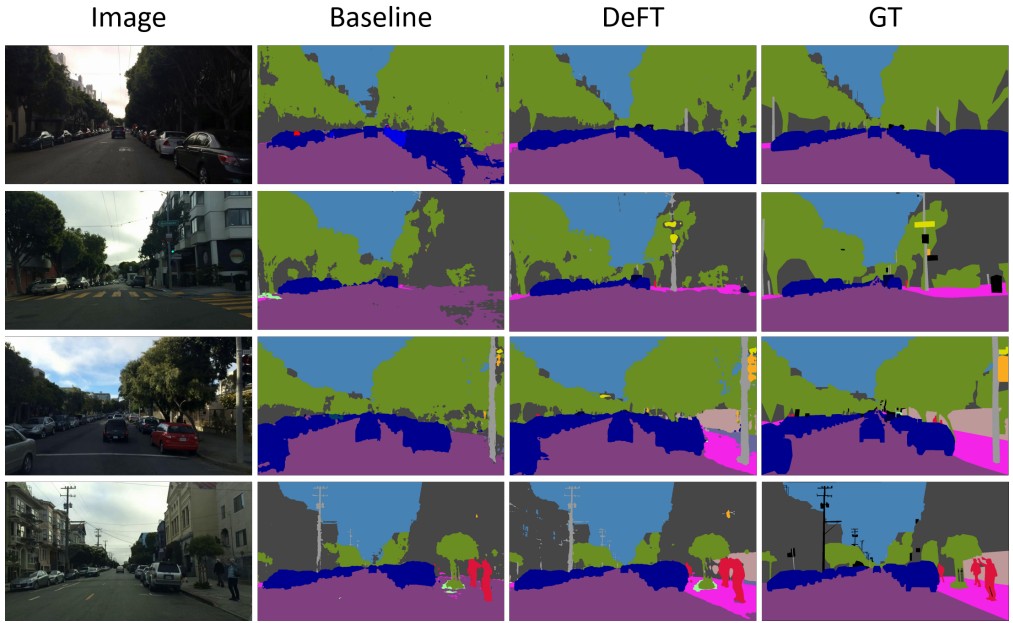

Figure 6: Qualitative results of DeFT and its baseline on BDD100K dataset.

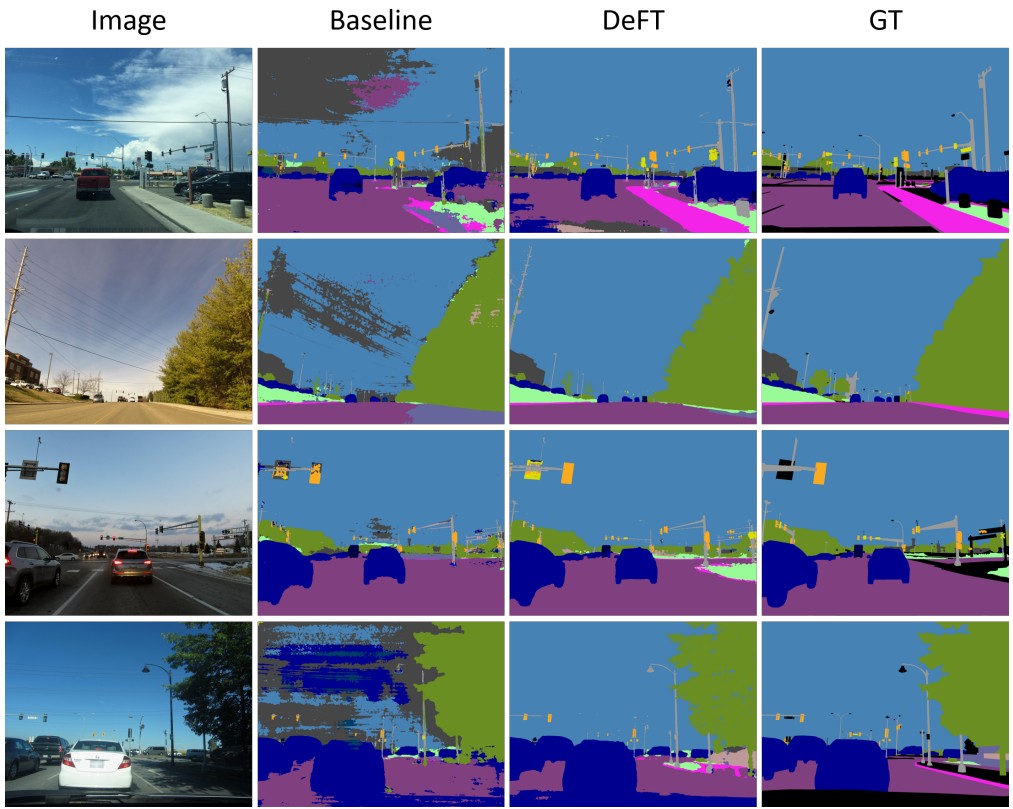

Figure 7: Qualitative results of DeFT and its baseline on Mapillary dataset.

| road | sidewalk | building | wall | fence |
|------|----------|----------|------|-------|
| pole | traffic light | traffic sign | vegetation | terrain |
| sky | person | rider | car | truck |
| bus | train | motorbike | bicycle | |

Figure 8: The color code of 19 classes on the training and test datasets.

