# OpenReview forum: "Decoupled Finetuning for Domain Generalizable Semantic Segmentation"
_ICLR.cc/2025/Conference — ICLR 2025 Poster_

### Official Review · Reviewer_ck24 · 2024-10-30

**Soundness:** 4
**Presentation:** 4
**Contribution:** 3
**Rating:** 6
**Confidence:** 4

**Summary:**

**Motivation**
The paper identifies a critical gap in the semantic segmentation field, where the standard practice of joint fine-tuning pre-trained encoders with randomly initialized decoders has not been adequately examined regarding its vulnerability to domain shift. This lack of investigation highlights a potential risk to domain generalization, motivating the need for a new approach to fine-tuning that can better address these challenges.

**Method**
The authors propose the Decoupled Fine-Tuning (DeFT) framework, which consists of two key stages:

Stage One: A frozen pre-trained encoder is used to preheat the decoder. This allows the decoder to learn task-relevant knowledge while keeping the encoder's generalization ability intact.
Stage Two: The fine-tuning process for the encoder and decoder is decoupled into two separate paths. Each path includes:
A Conventional Component (UC) that updates based on the loss gradient from the source domain.
A Generalization Component (GC) that updates through an exponential moving average biased towards its initialization, thus retaining its generalization capability.

**Contribution**
The introduction of the **DeFT** framework, which explicitly addresses the negative impact of joint fine-tuning on domain generalization in semantic segmentation tasks.
The provision of a novel approach to decouple the fine-tuning of encoders and decoders, leading to improved performance and robustness against domain shifts compared to existing methods that primarily focus on data or feature augmentation.

**Strengths:**

1. A novel training method, DeFT has been proposed, abandoning the previous joint fine-tuning approach and demonstrating its effectiveness through experiments.
2. Extensive experimental analyses have been conducted to validate the effectiveness of the experiments and assess the sensitivity to parameters, proving the efficacy of DeFT.
3. This method can be applied to multiple domains and provides ideas for future research.

**Weaknesses:**

1. I am worried that this method may significantly reduce computational speed because the introduction of two EMAs could have a considerable impact on training speed, even if it does not significantly affect inference speed.
2. There is a lack of insightful analysis regarding the performance loss associated with joint fine-tuning, making the motivation less compelling.

**Questions:**

1. I would like to know the impact of the two EMA updates in DeFT on training speed. Specifically, how much slower is it compared to the baseline? Could you provide detailed data on this?

2. In Table 8, I noticed that the EMA update ratio (β) has a significant impact on the experimental results, even with slight variations in this parameter. Could you provide a further analysis of the reasons behind this?

3. In Figure 1, the issue of target loss fluctuations is raised. Does DeFT address this problem? Could further experiments be conducted to investigate this?

---

> ### Author Response · Authors · 2024-11-24
>
> Thank you for the insightful and constructive comment! Below, we provide detailed responses to address your concerns.
>
> &nbsp;
>
> ### **Additional overhead in training time**
> To analyze additional computational and memory costs of DeFT, we measured the whole training time for 40K iterations and the peak GPU memory usage during the training. In both settings, the batch size is set to 4, and the cost is measured on GTAV with ResNet-50 backbone and only for the main training stage, except warm-up.
>
> DeFT imposes additional space-time complexity only in training, and the overhead is not significant: 12% increase in training time and 22% increase in memory footprint, as shown in Table R1. This is because (1) the number of learnable parameters is exactly the same as that of joint finetuning and (2) the training images or labels are shared between the two pathways. Since GC is not updated by the standard gradient-based backpropagation, there is no need to compute or store gradients for GC. Moreover, since the two pathways share training data at every iteration, we do not need to load training data twice. These allow DeFT not to simply “double” the training time or memory usage. With such a small computation and memory overhead in training, it achieves significant performance improvement. Last but not least, DeFT does not have any additional overhead in the inference process.
>
> &nbsp;
>
> |                           | Joint Finetuning | DeFT         |
> |---------------------------|-------------------|--------------|
> | Training Time (40K)       | 16h 02m          | 18h 00m      |
> | Peak GPU Memory Usage     | 13255.88 MB      | 16118.89 MB  |
>
> **Table R1.** Comparison for the training time and peak GPU memory usage between joint finetuning and DeFT.
>
> &nbsp;
>
> ### **Sensitivity for EMA update ratio $\beta$**
> While the differences between the EMA update rates in Table 8 look tiny, even such small variations lead to significantly different temporal ensemble results. For example, the EMA update rates 0.999 and 0.9999 lead to the ensemble coefficients for the initial state 4.1683e-18 and 0.0183, respectively, after 40K iterations. This gap causes a substantial performance difference in DeFT where GCs should stay close to their initial states.
>
> &nbsp;
>
> ### **Target loss fluctuation**
> We measured the target loss fluctuations on Cityscapes during training on GTAV for DeFT. As images cannot be attached to the comment, we have added the result as Figure 9 in the appendix of the revised version of our paper, and would be grateful if you refer to it. As shown in Figure 9, DeFT significantly mitigates the target loss fluctuation, exhibiting consistently low target loss during training.

---

> > ### Comment · Reviewer_ck24 · 2024-11-29
> >
> > Thank you for your reply. I have reviewed your feedback and adjusted my scores accordingly. While this approach does slightly increase the training time, the results suggest that the effort is worthwhile

---

> > > ### Author Response · Authors · 2024-11-29
> > > **Thank You!**
> > >
> > > Thank you for your quick response and for the positive feedback!
> > > We are happy to hear that your major concern has been addressed. If there are any remaining concerns that make you hesitant to increase your score from 6 (marginally above the acceptance threshold), please let us know. We are happy to respond!

---

### Official Review · Reviewer_wUh8 · 2024-11-01

**Soundness:** 2
**Presentation:** 2
**Contribution:** 3
**Rating:** 6
**Confidence:** 4

**Summary:**

The paper, titled Decoupled Finetuning for Domain Generalizable Semantic Segmentation, introduces a novel approach called DeFT. Traditional joint finetuning of an encoder and decoder often leads to overfitting to the source domain, which degrades generalization capabilities on unseen domains. To address this, DeFT proposes a two-stage framework: (1) warming up the decoder while keeping the pretrained encoder frozen to preserve its generalizable features, and (2) decoupling the finetuning of the encoder and decoder into two pathways, each involving a usual component (UC) and a generalized component (GC) updated by an exponential moving average. This decoupling helps retain initial generalizable knowledge while improving task-specific learning. The method significantly outperformed existing state-of-the-art domain generalization techniques across various domain shift scenarios.

**Strengths:**

1. Originality: DeFT introduces a unique decoupling strategy for finetuning the encoder and decoder, setting it apart from prior methods that only optimize them jointly. The method's use of an exponential moving average for GC updates further enhances its originality.
2. Quality: The paper is well-structured, providing comprehensive empirical evidence through experiments across five datasets. Ablation studies and comparisons with state-of-the-art methods reinforce the robustness of DeFT.
3. Clarity: The description of DeFT, supported by visual diagrams, pseudocode, and detailed explanations, is clear and facilitates understanding of the training process.

**Weaknesses:**

1. Complexity of Implementation: The decoupled pathway design and maintaining separate UC and GC pathways may introduce additional implementation complexity that is not addressed in terms of potential computational overhead. That is, there is no relevant model complexity analysis.
2. Sensitivity analysis experiments for the main parameters are not available.
3.The experimental equipment is not described in sec.4 EXPERIMENTS.

**Questions:**

1. How does the complexity of implementing DeFT, with its two separate pathways, impact training time and computational resources compared to standard joint finetuning methods?
2. Have the authors explored other update schemes for the GCs besides the exponential moving average? If so, what were the results, and how did they compare?

---

> ### Author Response · Authors · 2024-11-24
>
> Thank you for the insightful and constructive comment! Below, we provide detailed responses to address your concerns.
>
> &nbsp;
>
> ### **Complexity of Implementation**
> First of all, the code-level implementation of DeFT is fairly simple, as described in Section A of the appendix. To analyze additional computational and memory costs of DeFT, we measured the whole training time for 40K iterations and the peak GPU memory usage during the training. In both settings, the batch size is set to 4, and the cost is measured on GTAV with ResNet-50 backbone and only for the main training stage, except warm-up.
>
> DeFT imposes additional space-time complexity only in training, and the overhead is not significant: 12% increase in training time and 22% increase in memory footprint, as shown in Table R1. This is because (1) the number of learnable parameters is exactly the same as that of joint finetuning and (2) the training images or labels are shared between the two pathways. Since GC is not updated by the standard gradient-based backpropagation, there is no need to compute or store gradients for GC. Moreover, since the two pathways share training data at every iteration, we do not need to load training data twice. These allow DeFT not to simply “double” the training time or memory usage. With such a small computation and memory overhead in training, it achieves significant performance improvement. Last but not least, DeFT does not have any additional overhead in the inference process.
>
> &nbsp;
>
> |                           | Joint Finetuning | DeFT         |
> |---------------------------|-------------------|--------------|
> | Training Time (40K)       | 16h 02m          | 18h 00m      |
> | Peak GPU Memory Usage     | 13255.88 MB      | 16118.89 MB  |
>
> **Table R1.** Comparison for the training time and peak GPU memory usage between joint finetuning and DeFT.
>
> &nbsp;
>
> ### **Sensitivity analysis for the main parameters**
> In DeFT, the main parameters that have the biggest impact on performance are $\beta$ in Eq. (2) and $\alpha$ in Eq. (3).
>
> 1. **Regarding $\beta$**: The most important hyperparameter of DeFT is the EMA update rate $\beta$, which balances the pretrained knowledge and the task relevant knowledge of the GC. Results of the sensitivity analysis on this parameter were already reported in Table 8 of the paper.
>
> 2. **Regarding $\alpha$**: In the paper, we discuss the implicit regularization effect of DeFT on the distance from initialization, while also demonstrating that explicit regularization can be applied in conjunction. To this end, we employ a simple Euclidean distance-based regularization of Eq. (3), where the parameter $\alpha$ determines the contribution of the distance-based regularization. The impact of the explicit distance-based regularization has been already demonstrated in Table 3 of the paper. Please note that the explicit distance-based regularization is not the primary focus of our study; in the main tables of the paper comparing DeFT with prior work, the distance-based regularization has not been applied at all. So we did not provide an extensive ablation study on the impact of $\alpha$, but we are currently working on the analysis and will provide the results when they are ready.
>
> The other hyperparameters such as the learning rate and weight decay were set to the values used in the previous work.
>
> &nbsp;
>
> ### **Experimental equipment**
> We conducted our experiments using four RTX 3090 GPUs.
>
> &nbsp;
>
> ### **Other update schemes for GC**
> Great suggestion. We conducted additional experiments by replacing the update scheme for GC in DeFT (i.e., EMA) with various weight ensemble methods.
>
> &nbsp;
>
> (B) Simply averaging the latest UC weights with the initial weights, similar to WiSE-FT[1]
> - $\theta^\text{GC}_t = 0.5 * \theta^\text{UC}_t + 0.5 * \theta_0$
>
> (C) exponentially decreasing the later ensemble coefficient
> - $\theta^\text{GC}_t = (\alpha * \theta^\text{GC}\_{t-1} + \theta^\text{UC}_t)/T, \quad T = \sum\_{i=0}^{t-1} \alpha^i, \quad \alpha > 1$
>
> &nbsp;
>
> As shown in Table R2, the EMA biased towards initialization employed in DeFT clearly outperformed the other weight ensemble methods.
>
> &nbsp;
>
> |                       | Cityscapes | BDD100K | Mapillary | avg.  |
> |-----------------------|------------|---------|-----------|-------|
> | (A) **EMA (DeFT)**  | **50.06**|**43.17**|**50.51**|**47.91**|
> | (B) WiSE-FT           | 45.03      | 40.47   | 47.04     | 44.18 |
> | (C) Exp. Dec.         | 44.46      | 40.86   | 46.72     | 44.01 |
>
> **Table R2.**  Comparison between the EMA and other GC update schemes
>
> &nbsp;
>
> ### **References**
> [1] Mitchell Wortsman, Gabriel Ilharco, Jong Wook Kim, Mike Li, Simon Kornblith, Rebecca Roelofs, Raphael Gontijo Lopes, Hannaneh Hajishirzi, Ali Farhadi, Hongseok Namkoong, and Ludwig Schmidt. Robust fine-tuning of zero-shot models. In Proc. IEEE/CVF Conference on Computer Vision and Pattern Recognition (CVPR), 2022.

---

> ### Author Response · Authors · 2024-11-29
> **Gentle Reminder**
>
> Thank you once again for your thoughtful review! Your feedback has been incredibly helpful in improving our work.
>
> As the discussion period is nearing its end, we would be very grateful if you could take a moment to review the modifications and additional analysis we have made in response to your concerns.
> We hope these revisions address the points you raised.
>
> Once again, we sincerely appreciate your valuable time and effort in reviewing our work.

---

### Official Review · Reviewer_qfZ6 · 2024-11-03

**Soundness:** 3
**Presentation:** 2
**Contribution:** 4
**Rating:** 6
**Confidence:** 3

**Summary:**

This paper introduces Decoupled FineTuning (DeFT), a novel framework designed to improve domain generalization in semantic segmentation tasks, where models often perform poorly when exposed to domain shifts. The traditional joint fine-tuning of encoder and decoder networks can lead to overfitting on source domains, reducing generalizability to new, unseen domains.

The paper have three contributions:(1)By separating the fine-tuning of encoder and decoder components, DeFT avoids overfitting and improves the model's ability to generalize across domains.(2)Using EMA updates for GCs allows the model to retain initialization-based generalization properties, enhancing the robustness of learned representations.(3)DeFT is validated on five datasets (e.g., Cityscapes, BDD-100K, Mapillary), showing substantial improvements over state-of-the-art methods across diverse domain shift scenarios.

These contributions position DeFT as an effective framework for domain-generalizable semantic segmentation.

**Strengths:**

Originality:Rather than jointly fine-tuning encoder and decoder components, the paper innovates with a decoupled strategy. This approach uniquely assigns the encoder and decoder to two different components—Usual Components (UC) and Generalized Components (GC)—which are updated in parallel but through distinct pathways. This structure is particularly inventive in preventing overfitting to the source domain while preserving generalizable features,I think this is a very clever and useful method,At the same time, the author uses EMA to update GC and maintain its generalization ability in the fine-tuning process,This method is also very unique, different from the traditional method.

Quality:In this paper, five different data sets (such as Cityscapes, BDD-100K, GTAV, etc.) and a variety of domain offsets are comprehensively tested, and other methods (such as WildNet, SHADE, BlindNet, etc.) are compared.

Clarity:This paper clearly expounds each stage from the initialization of the decoder to the decoupling fine tuning, provides the pseudo-code of the algorithm, and effectively conveys the architecture and ablation experimental results by using diagrams and tables. I think this makes me clearly understand the whole algorithm process and the advantages of the algorithm.

Significance:This paper solves a key problem of domain generalization of semantic segmentation, and I think it plays a important role.

**Weaknesses:**

limitation:While the empirical results support the DeFT framework, the theoretical grounding behind the decoupling strategy (specifically the benefits of separating encoder and decoder updates into Usual and Generalized Components) is limited. Strengthening this theoretical component would help clarify why the decoupling approach reduces overfitting and improves generalization.

Suggestion:Consider providing a more detailed theoretical analysis or explanation, possibly referencing or building upon works in model fine-tuning and parameter space regularization.Your overview map is very brief, please enrich it if you can.

**Questions:**

1.Can you clarify why you chose specific datasets and domain shifts for evaluating DeFT?
2.Have you conducted ablation experiments on the updated method? For example, if you use EMA-based updates only for the encoder or only for the decoder, how will DeFT behave?

---

> ### Author Response · Authors · 2024-11-25
> **Official Comment by Authors (Part 1/2)**
>
> Thank you for the insightful and constructive comment! Below, we provide detailed responses to address your concerns.
>
> &nbsp;
>
> ### **More detailed theoretical explanation**
> Kumar et al. [1] showed that jointly finetuning a well-generalizable encoder with a randomly initialized decoder distorts the representation of the encoder, and proved this theoretically for the case of a two-layer linear neural network. To be specific, they demonstrated that such finetuning using only in-distribution (ID) data does not change the encoder's features for out-of-distribution (OOD) data although the encoder itself has been updated; this distortion deteriorates the encoder's generalization to OOD data. Further, such a distortion of one module also affects the other since loss gradients for updating the two are coupled; for example, the decoder's parameters are affected by the distortion of the encoder, which is referred to as "balancedness" in [2]. As a result, predictions of the entire model for OOD data could be corrupted. In other words, joint finetuning of the encoder and decoder on ID data (i.e., source domain data in domain generalization) distorts the generalizable knowledge of both modules, leading to poor generalization to OOD data (i.e., unseen target domain data in domain generalization).
>
> On the other hand, consider a linear probing case where the encoder is fixed and no feature distortion occurs. Lemma A.14. in [1] shows that the upper bound of OOD error of the linear probing is proportional to the difference between the pretrained encoder and the "optimal" encoder, which shows the lowest errors for both ID and OOD data. That is, in the linear probing case, a decoder coupled with more generalizable encoder results in a more generalizable final model. Based on this, we first decouple the encoder and decoder to prevent each module from being distorted by their jointly finetuned counterparts, coupling them with counterparts which are not affected by distribution (i.e., domain) shift thus are well generalizable.
>
> The optimization behavior from the perspective of individual network modules, such as encoders and decoders, as well as the interactions between these modules during the optimization process, seems to remain relatively underexplored. We expect that further foundational study in this direction might pave the way for more rigorous theoretical analysis of DeFT.
>
> &nbsp;
>
> ### **References**
> [1] Ananya Kumar, Aditi Raghunathan, Robbie Jones, Tengyu Ma, and Percy Liang. Fine-tuning can distort pretrained features and underperform out-of-distribution. In International Conference on Learning Representations
> (ICLR), 2022.
>
> [2] Simon S. Du, Wei Hu, and Jason D. Lee. Algorithmic regularization in learning deep homogeneous models: Layers are automatically balanced. In Proc. Neural Information Processing Systems (NeurIPS), 2018.

---

> ### Author Response · Authors · 2024-11-25
> **Official Comment by Authors (Part 2/2)**
>
> ### **Justification for the evaluation settings**
> Sorry for the lack of clear justification for the evaluation protocol, which should have been covered in the paper at the time of submission.
> First of all, we chose the datasets and domain shifts to compare DeFT with most of existing domain generalization (DG) methods in the same settings since they have been widely used as the standard evaluation protocol for DG in the literature.
>
> More importantly, the datasets and domain shifts enable us to simulate application scenarios that are intriguing and have great practical values. For example, {GTAV, SYNTHIA}→{Cityscapes, BDD100K, Mapillary} are synthetic-to-real generalization settings, and DG methods working in these settings may resolve the lack of labeled training data by exploiting synthetic data. Also, the Cityscapes→BDD100K setting can be used to evaluate a model’s robustness to distribution shifts in the wild by, for example, different geolocations and weather conditions.
>
> &nbsp;
>
> ### **Ablation experiments on the updated method**
> Thank you for the great suggestion! We conducted additional experiments on the two different update methods: using EMA-based updates (1) only for the encoder, and (2) only for the decoder. More specifically, we basically conducted joint finetuning of the encoder and decoder, yet replaced one module with its EMA version at each iteration, while keeping the other as is. These experiments were conducted on GTAV using ResNet-50 backbone while the EMA update rate $\beta$ was set to be 0.9999, the same as our DeFT. The result of these experiments is summarized in Table R1.
>
> &nbsp;
>
> | EMA (0.9999)        | Cityscapes | BDD100K | Mapillary | avg.  |
> |---------------------|------------|---------|-----------|-------|
> | only encoder        | 35.33      | 33.99   | 40.04     | 36.45 |
> | only decoder        | 44.79      | 40.35   | 46.47     | 43.87 |
> | **DeFT**           | **50.06** | **43.17** | **50.51** | **47.91** |
>
> **Table R1.** Ablation experiments on the two updated methods
>
> &nbsp;
>
> Please note that replacing a module with its EMA version at each iteration is equivalent to simply reducing the learning rate. Let $\theta_t$, $\theta'_t$ be a parameter and its EMA value at iteration $t$ respectively. Then, the $\theta_t$ and $\theta'_t$ are calculated as follows:
>
> $$
> \theta_t = \theta\_{t-1} - \lambda \cdot \Delta\_{\theta} L. \\
> $$
> $$
> \theta'_t = \beta \cdot \theta'\_{t-1} + (1 - \beta) \cdot \theta_t \\
> $$
> $$
> \theta_t := \theta'_t = \beta \cdot \theta\_{t-1}' + (1 - \beta) \cdot (\theta\_{t-1} - \lambda \cdot \Delta\_{\theta} L) \\
> = \beta \cdot \theta\_{t-1}' + (1 - \beta) \cdot \theta\_{t-1} - (1 - \beta) \cdot \lambda \cdot \Delta\_{\theta} L
> $$
>
> where $L$ is the loss calculated at iteration $t$ and $\lambda$ is a learning rate. However, $\theta'\_{t-1} = \theta\_{t-1}$, thus $\theta_t$ after the iteration $t$ is:
> $$
> \theta\_{t} = \theta\_{t-1} - (1 - \beta) \cdot \lambda \cdot \Delta\_{\theta} L
> $$
>
> &nbsp;
>
> Hope we correctly understand your intention. Otherwise, could you please provide more details about the updated method in the comment? We will gladly conduct additional experiments to resolve your concern more precisely!

---

> ### Author Response · Authors · 2024-11-29
> **Gentle Reminder**
>
> Thank you once again for your thoughtful review! Your feedback has been incredibly helpful in improving our work.
>
> As the discussion period is nearing its end, we would be very grateful if you could take a moment to review the modifications and additional analysis we have made in response to your concerns.
> We have also strengthened the theoretical foundation in Section B of the latest revision. We would greatly appreciate it if you could review it at your convenience.
> We hope these revisions address the points you raised.
>
> Once again, we sincerely appreciate your valuable time and effort in reviewing our work.

---

### Official Review · Reviewer_sHdo · 2024-11-03

**Soundness:** 3
**Presentation:** 3
**Contribution:** 3
**Rating:** 6
**Confidence:** 3

**Summary:**

In this work the authors propose a fine tuning strategy for encoder-decoder networks to improve domain generalization. The idea is to decouple the fine tuning of encoder and decoder by keeping a frozen copy of the decoder (encoder) when fine tuning the encoder (decoder), and then combine the encoder and decoder with their frozen copies via exponential moving averages. This seems to result in more robustness to overfitting and better generalization, showing improved performance in semantic segmentation scenarios.

**Strengths:**

- The idea is sound and well motivated.
- Comprehensive experiments that show a notable improvement in the generalization capabilities of the model

**Weaknesses:**

- The contribution of decoupling encoder and decoder learning, and the contribution of averaging with earlier frozen models, in my view, should be differentiated and ablated properly. In particular, I suggest to evaluate some baseline without decoupling but that averages the joint encoder-decoder model with earlier clones.
- The method is presented in an overcomplicated and confusing way, and be described in a simpler way. In particular the terminology usual component (UC) and generalized component (GC) to refer to the model and its frozen earlier clone is more confusing than clarifying.
- There is no theoretical analysis (even preliminary) of why the proposed approach improves generalization (the authors leave it to future work).
- There is no analysis of the additional computational and memory costs of the proposed approach.

**Questions:**

Please address the comments in the weakenesses section.

---

> ### Author Response · Authors · 2024-11-25
> **Official Comment by Authors (Part 1/2)**
>
> Thank you for the insightful and constructive comment! Below, we provide detailed responses to address your concerns.
>
> &nbsp;
>
> ### **An ablation study to observe the effectiveness of decoupled finetuning alone for DeFT**
> As you suggested, we designed and conducted an experiment to investigate the effect of decoupled finetuning and that of weight ensemble separately. To be specific, we jointly finetuned the encoder and decoder, and considered their EMA versions as the final model for evaluation. The experiment was conducted using ResNet-50 backbone and the model was trained on GTAV. The EMA update rate $\beta$ was set to 0.9999, the same as DeFT.
>
> As shown in Table R1, DeFT outperforms the aforementioned baseline (i.e., Joint + EMA). This result suggests that the proposed decoupled finetuning better preserves generalizable knowledge of the pretrained encoder and decoder than joint finetuning. Moreover, the result also shows  that the temporal weight ensemble yields a more generalizable model than those trained only by the joint finetuning, which empirically justifies the use of the temporal weight ensemble as the update scheme for GC in DeFT.
>
> &nbsp;
>
> |                           | Cityscapes | BDD100K | Mapillary | avg.  |
> |---------------------------|------------|---------|-----------|-------|
> | Joint Finetuning          | 42.32      | 40.33   | 44.88  | 42.51 |
> | Joint + EMA               | 48.30      | 42.29   | 49.02     | 46.54 |
> | **DeFT**                 | **50.06** |**43.17**|**50.51**|**47.91**|
>
> **Table R1.** An ablation study to isolate and observe the effects of decoupled fine-tuning without the influence of temporal weight ensembling

---

> ### Author Response · Authors · 2024-11-25
> **Official Comment by Authors (Part 2/2)**
>
> ### **Confusing writing, especially the terms GC and UC**
> We apologize for any confusion. We are continually striving to improve our writing. For better understanding, we have changed the terms Generalization Component (GC) and Usual Component (UC) to Retentive Component (RC) and Adaptive Component (AC), respectively. Please confirm these changes or let us know of any better alternatives, and we will revise the manuscript accordingly as soon as possible.
>
> &nbsp;
>
> ### **Theoretical foundation**
> Kumar et al. [1] showed that jointly finetuning a well-generalizable encoder with a randomly initialized decoder distorts the representation of the encoder, and proved this theoretically for the case of a two-layer linear neural network. To be specific, they demonstrated that such finetuning using only in-distribution (ID) data does not change the encoder's features for out-of-distribution (OOD) data although the encoder itself has been updated; this distortion deteriorates the encoder's generalization to OOD data. Further, such a distortion of one module also affects the other since loss gradients for updating the two are coupled; for example, the decoder's parameters are affected by the distortion of the encoder, which is referred to as "balancedness" in [2]. As a result, predictions of the entire model for OOD data could be corrupted. In other words, joint finetuning of the encoder and decoder on ID data (i.e., source domain data in domain generalization) distorts the generalizable knowledge of both modules, leading to poor generalization to OOD data (i.e., unseen target domain data in domain generalization).
>
> On the other hand, consider the linear probing case where the encoder is fixed and no feature distortion occurs. Lemma A.14. in [1] shows that the upper bound of OOD error of the linear probing is proportional to the difference between the pretrained encoder and the "optimal" encoder, which shows the lowest errors for both ID and OOD data. That is, in the linear probing case, a decoder coupled with more generalizable encoder results in a more generalizable final model. Based on this, we first decouple the encoder and decoder to prevent each module from being distorted by their jointly finetuned counterparts, coupling them with counterparts which are not affected by distribution (i.e., domain) shift thus are well generalizable.
>
> The optimization behavior from the perspective of individual network modules, such as encoders and decoders, as well as the interactions between these modules during the optimization process, seems to remain relatively underexplored. We expect that further foundational study in this direction might pave the way for more rigorous theoretical analysis of DeFT.
>
> &nbsp;
>
> ### **Additional computational and memory costs**
> To analyze additional computational and memory costs of DeFT, we measured the whole training time for 40K iterations and the peak GPU memory usage during the training. In both settings, the batch size is set to 4, and the cost is measured on GTAV with ResNet-50 backbone and only for the main training stage, except warm-up.
>
> DeFT imposes additional space-time complexity only in training, and the overhead is not significant: 12% increase in training time and 22% increase in memory footprint, as shown in Table R2. This is because (1) the number of learnable parameters is exactly the same as that of joint finetuning and (2) the training images or labels are shared between the two pathways. Since GC is not updated by the standard gradient-based backpropagation, there is no need to compute or store gradients for GC. Moreover, since the two pathways share training data at every iteration, we do not need to load training data twice. These allow DeFT not to simply “double” the training time or memory usage. With such a small computation and memory overhead in training, it achieves significant performance improvement. Last but not least, DeFT does not impose any additional overhead in the inference process.
>
> &nbsp;
>
> |                           | Joint Finetuning | DeFT         |
> |---------------------------|-------------------|--------------|
> | Training Time (40K)       | 16h 02m          | 18h 00m      |
> | Peak GPU Memory Usage     | 13255.88 MB      | 16118.89 MB  |
>
> **Table R2.** Comparison for the training time and peak GPU memory usage between joint finetuning and DeFT.
>
> &nbsp;
>
> ### **References**
> [1] Ananya Kumar, Aditi Raghunathan, Robbie Jones, Tengyu Ma, and Percy Liang. Fine-tuning can distort pretrained features and underperform out-of-distribution. In International Conference on Learning Representations
> (ICLR), 2022.
>
> [2] Simon S. Du, Wei Hu, and Jason D. Lee. Algorithmic regularization in learning deep homogeneous models: Layers are automatically balanced. In Proc. Neural Information Processing Systems (NeurIPS), 2018.

---

> ### Author Response · Authors · 2024-11-29
> **Gentle Reminder**
>
> Thank you once again for your thoughtful review! Your feedback has been incredibly helpful in improving our work.
>
> As the discussion period is nearing its end, we would be very grateful if you could take a moment to review the modifications and additional analysis we have made in response to your concerns. We have also strengthened the theoretical foundation in Section B of the latest revision. We would greatly appreciate it if you could review it at your convenience. We hope these revisions address the points you raised.
>
> Once again, we sincerely appreciate your valuable time and effort in reviewing our work.

---

> > ### Comment · Reviewer_sHdo · 2024-11-30
> >
> > I appreciate the authors' thorough response and effort. My main concerns are about efficiency, presentation and ablation were addressed and clarified. The preliminary theoretical analysis is also helpful. I updated my rating accordingly.

---

> > > ### Author Response · Authors · 2024-12-01
> > > **Thank You!**
> > >
> > > Thank you so much for your time, effort, and the positive feedback! We’re glad to hear that your main concerns have been addressed. If you have any other questions, please don’t hesitate to reach out—we’d be happy to help.

---

### Official Review · Reviewer_exsS · 2024-11-04

**Soundness:** 3
**Presentation:** 3
**Contribution:** 3
**Rating:** 6
**Confidence:** 4

**Summary:**

This paper investigates the vulnerability issue of joint finetuning, and proposes a novel finetuning framework called Decoupled FineTuning for domain generalization (DeFT) as a solution. DeFT operates in two stages. Its first stage warms up the decoder with the frozen, pretrained encoder so that the decoder learns task-relevant knowledge while the encoder preserves its generalizable features. In the second stage, it decouples finetuning of the encoder and decoder into two pathways, each of which concatenates a usual component (UC) and generalized component (GC); each of the encoder and decoder plays a different role between UC and GC in different pathways. DeFT significantly outperformed existing methods in various domain shift scenarios, and its performance could be further boosted by incorporating a simple distance regularization.

**Strengths:**

1. A  new finetuning framework dubbed Decoupled FineTuning (DeFT) is proposed for domain generalization. Overall, it is a simple and universal solution for DG tasks.

2. The experimental details are sufficient, the reproducibility is good, and the experimental results are convincing.

**Weaknesses:**

1. While the approach presented in this paper is quite straightforward, my primary concern is that its framework demonstrates effectiveness solely in practical applications. The authors provide an analysis based on parameter distance, yet they lack a more profound theoretical exploration. Given the standards of an ICLR paper, I believe a rigorous theoretical analysis is essential.

2. The paper exclusively utilizes convolutional backbones, such as ResNet-50, without any experiments on more advanced transformer-based backbones [1-2]. The absence of results with these state-of-the-art encoder-decoder frameworks limits the generalizability and relevance of the proposed method.

[1] HGFormer: Hierarchical Grouping Transformer for Domain Generalized Semantic Segmentation
[2] Kill Two Birds with One Stone: Domain Generalization for Semantic Segmentation via Network Pruning

**Questions:**

What are the performance comparisons on Transformer-based backbones?

A more rigorous theoretical analysis is essential.

---

> ### Author Response · Authors · 2024-11-25
>
> Thank you for the insightful and constructive comment! Below, we provide detailed responses to address your concerns.
>
> &nbsp;
>
> ### **Theoretical foundation**
> Kumar et al. [1] showed that jointly finetuning a well-generalizable encoder with a randomly initialized decoder distorts the representation of the encoder, and proved this theoretically for the case of a two-layer linear neural network. To be specific, they demonstrated that such finetuning using only in-distribution (ID) data does not change the encoder's features for out-of-distribution (OOD) data although the encoder itself has been updated; this distortion deteriorates the encoder's generalization to OOD data. Further, such a distortion of one module also affects the other since loss gradients for updating the two are coupled; for example, the decoder's parameters are affected by the distortion of the encoder, which is referred to as "balancedness" in [2]. As a result, predictions of the entire model for OOD data could be corrupted. In other words, joint finetuning of the encoder and decoder on ID data (i.e., source domain data in domain generalization) distorts the generalizable knowledge of both modules, leading to poor generalization to OOD data (i.e., unseen target domain data in domain generalization).
>
> On the other hand, consider a linear probing case where the encoder is fixed and no feature distortion occurs. Lemma A.14. in [1] shows that the upper bound of OOD error of the linear probing is proportional to the difference between the pretrained encoder and the "optimal" encoder, which shows the lowest errors for both ID and OOD data. That is, in the linear probing case, a decoder coupled with more generalizable encoder results in a more generalizable final model. Based on this, we first decouple the encoder and decoder to prevent each module from being distorted by their jointly finetuned counterparts, coupling them with counterparts which are not affected by distribution (i.e., domain) shift thus are well generalizable.
>
> The optimization behavior from the perspective of individual network modules, such as encoders and decoders, as well as the interactions between these modules during the optimization process, seems to remain relatively underexplored. We expect that further foundational study in this direction might pave the way for more rigorous theoretical analysis of DeFT.
>
> &nbsp;
>
> ### **Versaltility of DeFT at a transformer backbone**
> We evaluated DeFT incorporating  a transformer backbone, MiT-B5. As shown in Table R1, DeFT outperformed existing methods, such as DAFormer [3] and the combination of DAFormer and DGInStyle [4], using the same backbone. It is worth noting that DeFT achieved such outstanding performance without bells and whistles unlike DAFormer adapting its model architecture to domain generalization and DGInStyle relying heavily on extreme data augmentation; we strongly believe that employing such strategies will further improve the performance of DeFT. This result demonstrates that DeFT is a versatile training strategy applicable across various backbones.
>
> &nbsp;
>
> | SegFormer/MiT-B5          | Cityscapes | BDD100K | Mapillary | avg   |
> |---------------------------|------------|---------|-----------|-------|
> | DAFormer[3]       | 52.65      | 47.89   | 54.66     | 51.73 |
> | DAFormer + DGInStyle[4] | 55.31      | 50.82   | 56.62     | 54.25 |
> | **DeFT (Ours)**  | **57.16**      | **49.32**   | **59.99**     | **55.49** |
>
> **Table R1.** Comparison between DeFT and other methods incorporating MiT-B5 backbone. All models are trained on GTAV and evaluated on Cityscapes, BDD100K, and Mapillary.
>
> &nbsp;
>
> ### **References**
> [1] Ananya Kumar, Aditi Raghunathan, Robbie Jones, Tengyu Ma, and Percy Liang. Fine-tuning can distort pretrained features and underperform out-of-distribution. In International Conference on Learning Representations
> (ICLR), 2022.
>
> [2] Simon S. Du, Wei Hu, and Jason D. Lee. Algorithmic regularization in learning deep homogeneous models: Layers are automatically balanced. In Proc. Neural Information Processing Systems (NeurIPS), 2018.
>
> [3] Lukas Hoyer, Dengxin Dai, and Luc Van Gool. Daformer: Improving network architectures and training strategies for domain-adaptive semantic segmentation. In Proc. IEEE/CVF conference on computer vision and pattern recognition (CVPR), 2022.
>
> [4] Yuru Jia, Lukas Hoyer, Shengyu Huang, Tianfu Wang, Luc Van Gool, Konrad Schindler, and Anton Obukhov. Dginstyle: Domain-generalizable semantic segmentation with image diffusion models and stylized semantic control. In Proc. European Conference on Computer Vision (ECCV), 2024.

---

> > ### Comment · Reviewer_exsS · 2024-11-28
> >
> > I appreciate the authors' efforts in responding to my concerns. But in rebuttal, the comparison table is too simple, and the theoretical proof is only descriptive. I hope the author can include these responses in the revised draft, including sufficient comparisons with more Transform-based approaches [1-4] and theoretical analysis. I will upgrade my rating based on the revised draft.
> >
> > [1] Daformer: Improving network architectures and training strategies for domain-adaptive semantic segmentation.
> > [2] Dginstyle: Domain-generalizable semantic segmentation with image diffusion models and stylized semantic control.
> > [3] Kill Two Birds with One Stone: Domain Generalization for Semantic Segmentation via Network Pruning.
> > [4] HGFormer: Hierarchical Grouping Transformer for Domain Generalized Semantic Segmentation.

---

> > > ### Author Response · Authors · 2024-11-28
> > >
> > > Thank you for your prompt feedback!
> > >
> > > In the latest revision, we strengthened the theoretical foundation of DeFT (Section B), cited and discussed all the four aforementioned papers [1-4] (Section 2), and added the preliminary results of applying DeFT to transformer-based approaches [1,2] (Section E); the results in Section E of the Appendix will be moved to the main paper after a more thorough revision. We are still working on integrating DeFT with HGFormer [4], and have requested the official codebase of [3] from the authors; results of these experiments will be reported here as soon as they are available.

---

> ### Comment · Reviewer_exsS · 2024-11-29
>
> Thanks for offering the latest manuscript. I have checked it. On several points of my concern, the article has certainly been improved. So I will raise my score to 6. Although the idea is straightforward, I believe this approach will become a new paradigm for training in the DG field in the future.

---

> > ### Author Response · Authors · 2024-11-29
> > **Thank You!**
> >
> > Thank you very much for your time and effort and for the positive feedback! If you have any remaining concerns, please let us know and we'll be happy to respond further.

---

### Author Response · Authors · 2024-11-24
**Common response**

We sincerely appreciate the detailed and constructive feedback from all the reviewers. We are particularly grateful for the recognition of our work's novelty (all), extensive experiments (all), applicability to different domains (ck24), and clarity (wUh8, qfz6). Before addressing individual reviews, we would like to briefly outline the key points of our rebuttal:

### **Additional Analysis**
- Analysis on computational overhead introduced by DeFT.
- Analysis on the target loss fluctation of DeFT on Cityscapes dataset.
- Theoretical foundation of DeFT.

### **Extended Evaluation:**
- Additional evaluation using a transformer backbone.
- Additional ablation study on the effectiveness of the decoupling strategy.
- Additional experiments with two update methods: EMA-based updates applied exclusively to either the encoder or the decoder.
- Additional evaluations using different weight ensemble methods.

Please note that we are continuing with additional experiments and will update our progress and responses as soon as they are completed.

---

### Author Response · Authors · 2024-11-28
**Updated Revision**

Dear reviewers,

We sincerely appreciate your thoughtful and valuable feedback during the review process.

We have uploaded the revised version of our paper, incorporating the following modifications:
- Additional citations and discussions of the papers aformentioned by reviewer exsS (Section 2.1)
- More detailed theoretical foundation of DeFT (Section B of the Appendix)
- An analysis on the target loss fluctuation (Section D of the Appendix)
- Additional experiments on transformer backbone (Section E of the Appendix)

Thank you once again for your time and effort in providing insightful comments on our work.

---

### Meta-Review · Area_Chair_2c8z · 2024-12-22

**Metareview:**

Dear authors,

Thank you for submitting draft. This draft has received unanimous rating of 6 (marginally above acceptance level). One of the concern by the reviewers is "theoretical foundation", although factored during the feedback period, we encourage authors to further solidify it in the camera ready version. Please take into account comments and suggestions of all the authors.  We believe that readability of the draft could be further improved.

Although, acceptance is not contingent over it, but we will encourage authors to release code.


regards

AC

**Additional Comments On Reviewer Discussion:**

sHdo, exsS and ck24 increased their ratings. ck24 indicated that score has been adjusted.
One concerns by the reviewer was "theoretical aspect", which authors have replied to. However, it could be further improved.
During the discussion following points were noted.
*exsS: this approach will become a new paradigm for training in the DG field in the future.
*ck24 believes that methods increased complexity and challenging implementation "significantly diminish its practical value" and thus limits its broader applicability. However, ck24 did state method garners good results.
* sHdo: "vote for acceptance because the approach seems valuable in practice and the authors addressed properly my main concerns in the rebuttal."

My main inclination towards acceptance is based on something that appears from reviewers comments: method is simple and it seems to works (although one reviewer is concerned of how diverse application of it could be).

---

### Decision · Program_Chairs · 2025-01-22

Accept (Poster)